Analysis of microbial diversity and functions in sediments and overlying water of the Shiliu River

Li Yazi 1
Zhang Shuhong 1
Guo Yumei 2
Xu Ke 1
Zhang Xiqing 3
Pan Mengfei 3
Sun Qiaoping 4
Zhang Yanfang 4
Fan Yongshan 1 fanyongshan@126.com
1 Department of Life Sciences, Tangshan Normal University , Tangshan, Hebei , China
2 Institute for Microbiological Examination, Shijiazhuang Center for Disease Control and Prevention , Shijiazhuang, Hebei , China
3 College of Life Sciences, Agricultural University of Hebei , Baoding, Hebei , China
4 Hebei Green Dream Environmental Science & Technology Co., Ltd , Tangshan, Hebei , China
Yapıcı Sercan
Electronic publication date: 2025 Aug 29
Publication date: 2025
Volume: 13
Electronic Location ID: e19979
Received 2025 May 7; Accepted 2025 Aug 1
Copyright: © 2025 Li et al.
Copyright year: 2025
Copyright holder: Li et al.
License: This is an open access article distributed under the terms of the Creative Commons Attribution License, which permits unrestricted use, distribution, reproduction and adaptation in any medium and for any purpose provided that it is properly attributed. For attribution, the original author(s), title, publication source (PeerJ) and either DOI or URL of the article must be cited.
License URL: https://creativecommons.org/licenses/by/4.0/

Keywords: Shiliu river, Sediments, Overlying water, Diversity, Functional prediction

Funding: Science Research Project of Hebei Education Department QN2025283 Scientific Research Foundation of Tangshan Normal University 20256129068, 2024HX22 Central Government Guides Local Funds for Science and Technology Development 246Z3610G This study was financially funded by the Science Research Project of Hebei Education Department (QN2025283); the Scientific Research Foundation of Tangshan Normal University (20256129068, 2024HX22); and the Central Government Guides Local Funds for Science and Technology Development (246Z3610G). The funders had no role in study design, data collection and analysis, decision to publish, or preparation of the manuscript.

==============================
Background

With the acceleration of urbanization, urban rivers have become a significant component of the urban ecosystem, attracting considerable attention regarding their ecological status and biodiversity. This study focuses on the Shiliu River, aiming to analyze the microbial diversity and functions present in the overlying water and sediments of severely polluted areas.

Methods

This study investigated the Shiliu River. In August 2024, sediment and overlying water samples were collected from its severely polluted reaches. The NextSeq 2000 PE300 platform was employed for sequencing to detect bacterial and fungal taxa abundances. PICRUSt and FUNGuild predicted sample functional abundances using bacterial 16S rRNA and fungal internal transcribed spacer (ITS) gene sequences, respectively.

Results

The findings demonstrate that sediments exhibit higher bacterial and fungal richness than overlying water, with significant discrepancies in bacterial and fungal community compositions. Dominant taxa differ at both phylum and genus levels: in sediments, the predominant bacterial phylum is Proteobacteria and genus norank_Anaerolineaceae, while the dominant fungal phylum is Rozellomycota and genus unclassified_Rozellomycota. In overlying water, the bacterial phylum remains Proteobacteria but the dominant genus shifts to Acinetobacter, whereas fungal phyla and genera (Rozellomycota and unclassified_Rozellomycota) are consistent with sediments. Kyoto Encyclopedia of Genes and Genomes (KEGG) functional annotation identifies 25 metabolic pathways, with amino acid metabolism-related genes showing the highest abundance in both environments. Clusters of Orthologous Genes (COG) annotation reveals the highest abundance of [R] General function prediction in both sample groups, and FUNGuild analysis indicates that Animal Endosymbiont-Animal Pathogen-Plant Pathogen-Undefined Saprotroph is the most prevalent functional category in both sediments and overlying water. This study provides a microbiological foundation by clarifying microbial community structures (dominant phyla, functional taxa), decoding pollutant-degrading metabolic potentials (N/C cycling pathways), and identifying river health ecological indicators. This enables targeted bioremediation strategies (e.g., sediment microbial consortia for nutrient removal) and integrates microbial ecological data into urban river restoration.

Conclusions

This study reveals the microbial community structures in the sediments and overlying water of the polluted Shiliu River, finding diverse patterns with higher richness in sediments, Proteobacteria and Ascomycota as dominants. Shared taxa have different abundances, indicating niche differentiation. Sediments have enriched nitrogen/carbon cycling pathways for pollutant degradation. These results offer a microbiological basis for urban river restoration, identify bioremediation-target taxa, and stress the integration of microbial ecology into pollution management.

Introduction

Rivers are a vital component of the ecosystem, playing a crucial role in maintaining regional ecological balance, providing water resources, recreation and supporting biodiversity (Miao et al., 2022; Xu et al., 2023; Xie et al., 2021). Microorganisms, as essential members of the river ecosystem, are indispensable in various ecological processes, including material cycling, energy flow, pollutant degradation, nutrient recycling, bioremediation potential, and serving as ecological indicators (Borton et al., 2025; Wei et al., 2024). The microbial communities present in the sediments and overlying water not only reflect the health of the river ecosystem but also are crucial for the proper functioning of its ecological processes. The Shiliu River in Tangshan City, as a significant local water body, has ecological environment directly influencing the ecological security of the surrounding areas and the quality of life for residents. Industrial enterprises discharging wastewater into the river (e.g., heavy metals like hexavalent chromium, organic pollutants, and suspended solids) contaminate farmland soil and crops. Untreated industrial sewage containing hexavalent chromium and organic substances directly enters the river, potentially causing soil degradation and reducing crop quality through bioaccumulation (Liu & Xue, 2003). With the acceleration of urbanization and the rise in industrial activities, the Shiliu River faces a series of environmental challenges, including water pollution and ecological degradation (Carpenter et al., 1998). Understanding the microbial diversity and functions of the sediments and overlying water in the Shiliu River is vital for elucidating the operational mechanisms of the river ecosystem, evaluating its ecological health status, and formulating targeted ecological restoration strategies.

Microbial diversity serves as the foundation for the stability and functionality of ecosystems (Zhai et al., 2024). Analyzing microbial diversity allows for a better understanding of ecosystems’ responses to environmental changes (Zeglin, 2015; Amend et al., 2016). For instance, in polluted water bodies, the structure of microbial communities may change significantly, such as sensitive taxa may decline, while taxa with specific pollutant-degradation capabilities may increase (Sun et al., 2024; Peng et al., 2024). Furthermore, the functions of microorganisms, particularly those involved in the cycling of essential elements such as carbon, nitrogen, and phosphorus, are crucial for the normal operation of river ecosystems (Clark et al., 2022). Nitrifying and denitrifying bacteria, for example, play a vital role in the nitrogen cycle by converting ammonia-nitrogen into nitrate-nitrogen or nitrogen gas, thereby influencing the content and form of nitrogen in water bodies (Zumft, 1997). Currently, research on the Shiliu River primarily focuses on the analysis of physicochemical water quality indicators, with relatively few studies addressing its microbial diversity and functions. This study aims to conduct a comprehensive analysis of the diversity characteristics of microorganisms in the overlying water and sediments of the Shiliu River in Tangshan City, exploring their potential functions in driving Kyoto Encyclopedia of Genes and Genomes (KEGG)-annotated metabolic pathways (e.g., nitrogen/carbon cycles mediated by nirS and nosZ genes) and providing a scientific basis for ecosystem protection and restoration. Beyond this, the research uncovers microbial community indicators (e.g., Proteobacteria and Rozellomycota) for ecological health assessment, and establishes a basis for isolating indigenous denitrifying strains, developing pollution detection biomarkers, and promoting interdisciplinary applications in ecological bioremediation.

Materials and Methods

General situation of the study area and sample collection

The Shiliu River originates from the hilly and mountainous regions in the northern part of the Guye District in Tangshan City, Hebei Province, China. Its upstream encompasses the majority of Guye District and traverses six townships, namely Shuiyu Township, Dazhuangtuo Township, and Xijiatao Township within Guye District, as well as Wali Town, Kaiping Town, and Yuehe Town in Kaiping District. The Shiliu River converges into the Douhe River at Wangpanzhuang Town in the Kaiping District, serving as a significant tributary of the Douhe River system in Tangshan, HeBei, China. Historically, the Shiliu River faced severe pollution issues, with certain sections exhibiting dirty and foul-smelling water, accompanied by oil stains and debris floating on the surface. The primary sources of this pollution included dewatering effluents from nearby coal mines and domestic sewage. In August 2024, sediments samples (Group B: B1, B2, B3) and overlying water samples (Group D: D1, D2, D3) were collected from the heavily polluted sections of the Shiliu River (He & Zhang, 2018). At each sampling point, three overlying water samples of 2 L with uniform mixing were gathered. Sediments were collected using a sterile sampler spatula at three designated sites (Group B: B1, B2, B3) in Shiliu River. Surface sediments (0–10 cm depth) were carefully scooped to avoid disturbance, transferred into sterile 50 mL centrifuge tubes, preserved with ice packs at low temperature, and promptly transported back to the laboratory.

DNA extraction and high-throughput sequencing

The E.Z.N.A™ Mag-Bind Soil DNA Kit (Omega Bio-tek, Norcross, GA, U.S.) was used to extract genomic DNA from the samples. The highly variable V3-V4 region of the bacterial 16S rRNA gene was amplified by PCR with the universal primers 314F (5′-CCTAYGGGRBGCASCAG-3′) and 806R (5′-GGACTACNNGGGTATCTAAT-3′). The ITS3-ITS4 region of fungi was amplified by PCR with the universal primers ITS3 (5-GATGAAGAACGYAGYRAA-3′) and ITS4 (5′-TCCTCCGCTTATTGATATGC-3′) (Ni et al., 2025; Huang et al., 2021a). Agarose gel electrophoresis was used to detect the purity of PCR samples and the size of the library. The Qubit 3.0 fluorometer was used to determine the library concentration. All samples were pooled at equal mass ratios. The amplification products were sent to Sangon Biotech (Shanghai) Co., Ltd. (Shanghai, China) for sequencing using the nextseq2000 PE300 sequencing platform. Each sample had three replicates.

Data processing

Data processing of microbial samples from sediments and overlying water: After obtaining sequencing data for all samples, the cutadapt 1.1.8 software was used to identify and remove the primer adapter sequences. Subsequently, PEAR0.9.8 was utilized to splice (merge) paired-end reads into a single sequence based on the overlap relationship between them. The spliced paired reads were then distinguished and identified according to the barcode tag sequences for obtain the data of each sample. Following this, PRINSEQ 0.20.4 was used to excise bases with a quality value below 20 at the tail of the reads, thereby ensuring the validity of the data for each sample. The Silva_v138.1 database facilitated the comparison of the measured 16S rRNA sequences, while the UNITE_v9.0 database was used for comparing the measured internal transcribed spacer (ITS) sequences, allowing for taxa classification and analysis. The software USEARCH 11.0.667 generated the taxa abundance table at various taxonomic levels. The barplot function in R version 3.6.0 (R Core Team, 2019) were then employed to visualize the community structure diagrams of the samples at each level. The PCA plot was generated using the ggscatter function from the ggpubr package in R v3.6.0, and the UPGMA dendrogram was constructed using the hclust function in R v3.6.0 combined with the ape package. Additionally, Mothur 1.43.0 was utilized to evaluate the alpha diversity index of the samples (Schloss et al., 2009). To predict the functional abundance of the samples, PICRUSt 2 2.5.2 was used based on the abundance of 16S rRNA gene sequences in bacterial samples. The STAMP software was utilized to annotate 16S rRNA gene sequences derived from sediment and overlying water bacterial samples collected from the Shiliu River, referencing the KEGG and Clusters of Orthologous Genes (COG) databases. Additionally, FUNGuild 1.0 software was employed to annotate ITS gene sequences from sediment and overlying water fungal samples of the Shiliu River, relying on published literature and data from authoritative websites.

Statistical analysis

The Mothur 1.43.0 software was used to calculate the alpha diversity indices of the samples, including the observed taxa (ACE), Chao1 index, Simpson index, and Shannon index (Schloss et al., 2009). Excel 2010 was used to analyze and process the data and create corresponding tables. The Graphpad 5.0 software was used for T-test analysis.

Results

Statistics of sample sequencing data

Following the sequencing of bacteria and fungi in the sediments and overlying water, the detection results for each sample were obtained through barcode identification (GenBank: PRJNA1251787). A total of 170,775 valid sequences were retrieved from the sediment bacterial samples, yielding an average of 56,925 valid sequences per sample. In contrast, the overlying water bacterial samples produced a total of 181,579 valid sequences, with an average of 60,526 valid sequences per sample (Table S1). At a 97% similarity threshold, sediment bacterial samples yielded 110,225 OTUs, and overlying water bacterial samples contained 149,925 OTUs (Table S1). Notably, the number of unique OTUs in sediment bacterial samples exceeded that in the overlying water. Only 1,381 OTUs were shared between sediment and water (Fig. 1A), confirming distinct microbial species compositions.

Figure 1 Venn diagrams of the distribution of bacteria and fungi in sediment samples (B) and overlying water samples (D).

(A) Venn diagram of bacterial OTUs, where the numbers indicate the total OTUs in sediments (3,909), overlying water (2,139), and their shared OTUs (1,381); (B) Venn diagram of fungal OTUs, where the numbers indicate the total OTUs in sediments (1,398), overlying water (621), and their shared OTUs (526).

A total of 475,811 valid sequences were obtained from the sediment fungal samples (GenBank: PRJNA1253430). Each sample generated a minimum of 122,682 valid sequences, with an average of 158,604 valid sequences. In contrast, 385,430 valid sequences were obtained from the overlying water fungal samples, where each sample produced at least 107,075 valid sequences, averaging 128,477 valid sequences (Table S2). At a 97% similarity threshold, sediment fungal samples contained 237,763 operational taxonomic units (OTUs) and overlying water fungal samples yielded 100,958 OTUs (Table S2). The number of unique OTUs in sediment fungal samples exceeded that in the overlying water, with only 526 OTUs shared between the two (Fig. 1B). This confirms distinct microbial species compositions between sediment and water.

Alpha diversity analysis

Alpha diversity is mainly reflected by six indices, namely Simpson, Chao1, ACE, Shannon, Shannoneven, and Coverage, which reflect richness and evenness (Table S3). Among the sediment bacterial samples, all six indices were significantly higher than those in the overlying water (P < 0.05). This indicates that the richness in sediment bacterial samples is higher than that in the overlying water. The rarefaction curves further confirm this view (Fig. 2A). In addition, the Coverage index of each sample in this study was 1 or close to 1, indicating that the sequencing results are reliable (Table 1).

Figure 2 Rarefaction curves of bacteria and fungi from sediment samples (B1, B2, B3) and overlying water samples (D1, D2, D3).

(A) Rarefaction curve of bacteria; (B) rarefaction curve of fungi.

Table 1 The average value and standard deviation of the bacterial-diversity indices in sediment and overlying water samples.

Index	Sample ID	Shannon	Chao1	Ace	Simpson	Shannoneven	Coverage	
Average ± SD	B	6.50 ± 0.01	2,952.20 ± 95.25	3,068.15 ± 101.87	0.01 ± 0.00	0.82 ± 0.00	0.99 ± 0.00	
Average ± SD	D	4.76 ± 0.19	1,502.10 ± 185.57	1,764.50 ± 10.39	0.03 ± 0.00	0.65 ± 0.01	0.99 ± 0.00	

Among the sediment fungal samples, all six diversity indices were higher than those in the overlying water (Table S4). Specifically, the Chao1 and Ace indices of the sediment fungal samples were significantly higher than those in the overlying water (P < 0.05). This indicates that the richness of the sediment fungal samples is higher than that in the overlying water. The dilution curves further confirm this conclusion (Fig. 2B). In addition, the coverage index of the fungal samples was 1 or close to 1, indicating that the sequencing results are reliable (Table 2).

Table 2 The average value and standard deviation of the fungal-diversity indices in sediment and overlying water samples.

Index	Sample ID	Shannon	Chao1	Ace	Simpson	Shannoneven	Coverage	
Average ± SD	B	6.50 ± 0.01	2,952.20 ± 95.25	3,068.15 ± 101.87	0.01 ± 0.00	0.82 ± 0.00	0.99 ± 0.00	
Average ± SD	D	4.76 ± 0.19	1,502.10 ± 185.57	1,764.50 ± 10.39	0.03 ± 0.00	0.65 ± 0.01	0.99 ± 0.00	

Beta diversity analysis

Principal components analysis (PCA) was conducted using Euclidean distance was performed to visualize community structure, with PC1 explaining 80.76% and 68.66% of the variance for bacterial and fungal samples, respectively. Differences among sample groups were evaluated through PCA. In the results, distinct colors were used to represent different groups. The proximity of samples indicates similarity in microbial composition; conversely, greater distances reflect more significant differences. Notably, sediment bacterial and fungal samples were distinctly separated from the overlying water, indicating substantial differences in bacterial and fungal types (Fig. 3).

Figure 3 PCA diagrams of bacterial and fungal from sediment samples (B) and overlying water samples (D); (A) principal component analysis of bacterial; (B) principal component analysis of fungal.

Unweighted pair group method with arithmetic mean (UPGMA) cluster analysis results also fully corroborate the above results. In UPGMA, branches of different colors represent different groups. UPGMA clustering based on Bray-Curtis distance showed distinct clustering of sediment (B1-B3) and overlying water (D1-D3) samples, with branch lengths >0.1 separating the two groups, confirming significant community differentiation (Fig. 4).

Figure 4 UPGMA clustering analysis of bacteria and fungi in sediment samples (B1, B2, B3) and overlying water samples (D1, D2, D3).

(A) UPGMA clustering analysis of bacterial; (B) UPGMA clustering analysis of fungal.

Taxonomic composition analysis

Microbial community composition at the phylum level

At the phylum level, fourteen dominant phyla were identified through the phylum-level composition analysis of sediment and overlying water bacterial samples. The dominant phyla in sediment samples included Proteobacteria (32.38 ± 2.85%), Chloroflexi (20.09 ± 4.59%), and Bacteroidota (9.44 ± 0.46%). In the bacterial samples from overlying water, the predominant phyla were Proteobacteria (56.90 ± 12.70%), Actinobacteriota (17.07 ± 3.75%), and Planctomycetota (11.44 ± 3.32%) (Fig. 5A).

Figure 5 Bar chart of the relative abundances of bacteria and fungi at the phylum level in sediment samples (B1, B2, B3) and overlying water samples (D1, D2, D3).

(A) The relative abundances of bacteria at the phylum level; (B) the relative abundances of fungi at the phylum level.

Nine dominant phyla were selected based on a phylum-level composition analysis of fungal samples from sediment and overlying water. The dominant phyla included Rozellomycota (69.34 ± 4.71%), Ascomycota (11.06 ± 2.15%), and unclassified_Fungi (8.41 ± 1.42%). In contrast, the dominant phyla in the overlying water samples (D1, D2, D3) were Rozellomycota (46.63 ± 21.75%), unclassified_Fungi (30.37 ± 5.65%), and Ascomycota (12.80 ± 11.35%) (Fig. 5B).

Microbial community composition at the genus level

At the genus level, 49 dominant genera were identified from the bacterial genus-level composition of sediment and overlying water samples. Genus with an abundance proportion of less than 1% across all samples were categorized as “Others”. Notably, the dominant bacterial genera in sediment samples included norank_Anaerolineaceae (6.32 ± 1.80%) and norank_Bacteroidetes_vadinHA17 (4.37 ± 0.26%). In contrast, the dominant genera in the overlying water bacterial samples were Acinetobacter (11.07 ± 2.22%) and CL500-29_marine_group (6.79 ± 0.79%) (Fig. 6A).

Figure 6 Bar chart of the relative abundances of bacteria and fungi at the genera level in sediment samples (B1, B2, B3) and overlying water samples (D1, D2, D3).

(A) Bar chart of the relative abundances of bacteria at the genera level; (B) bar chart of the relative abundances of fungi at the genera level.

Following the analysis of the genus-level composition of sediment and overlying water fungi, 15 dominant genus were identified. Genus with an abundance proportion of less than 1% across all samples were categorized as “Others”. Among these, the dominant fungal genera in sediment samples were unclassified_Rozellomycota (54.41 ± 4.63%) and unclassified_Branch03 (9.46 ± 0.63%). In the overlying water samples, the dominant fungal genera were unclassified_Fungi (30.37 ± 5.65%) and unclassified_Rozellomycota (27.16 ± 13.18%) (Fig. 6B).

Functional gene prediction analysis

KEGG function prediction analysis

The analysis focused on the 25 KEGG metabolic pathways with the lowest P-values (Welch’s t-test, P < 0.05) in bacterial samples from sediment and overlying water, representing the most significantly differentially abundant functions between habitats. The results from KEGG functional annotation indicated that the most abundant functional gene across both sample types was associated with amino acid metabolism (Fig. 7A). Notably, the pathways with red (P < 0.05) in Fig. 7A, such as Xenobiotics biodegradation and metabolism, exhibited significant inter-habitat differences (Table S3). Furthermore, this study identified 24 functional genes related to nitrogen metabolism (Fig. 8). The nitrogen metabolism processes detected at the study included nitrification, assimilatory nitrate reduction, dissimilatory nitrate reduction, and nitrogen fixation. These processes may be linked to the in adequate treatment or collection of domestic sewage discharged by local residents.

Figure 7 Functional gene prediction analysis of bacteria in sediment samples (B) and overlying water samples (D).

(A) KEGG function prediction a nalysis of bacteria; (B) COG function prediction analysis. The fig on the left shows the abundance ratios of different functions in the two groups of samples. The middle part shows the difference ratios of functional abundances within the 95% confidence interval. The values on the far right are P-values. P-value less than 0.05 indicates a significant difference and is marked in red.

Figure 8 Heat map of functional genes related to nitrogen metabolism in sediment samples (B1, B2, B3) and overlying water samples (D1, D2, D3).

COG function prediction analysis

The 24 COG metabolic pathways exhibiting the lowest P-values in the sediment and overlying water bacterial samples were analyzed (Fig. 7B). COG annotation showed that “[R] General function prediction” had the highest abundance in both groups, while functions marked in red (P < 0.05) in Fig. 7B—such as “[S] Function unknown”—exhibited significant inter-habitat differences (Table S4).

FUNGuild functional prediction analysis

The results indicated that the predominant functional guilds in the sediment were “Animal Endosymbiont-Animal Pathogen-Plant Pathogen-Undefined Saprotroph” (1.94%), “Undefined Saprotroph (0.99%), and “Insect Parasite-Undefined Saprotroph” (0.47%). In the overlying water, the main functional guilds included “Animal Endosymbiont-Animal Pathogen-Plant Pathogen-Undefined Saprotroph” (3.06%), “Undefined Saprotroph” (2.36%), and “Endophyte-Plant Pathogen” (1.13%) (Fig. 9) (Table S5).

Figure 9 Bar chart of the statistical results of FUNGuild functional classification of fungi in sediment samples (B1, B2, B3) and overlying water samples (D1, D2, D3).

The horizontal axis represents the samples, and the vertical axis represents the abundance proportion of the guild in different samples.

Discussion

Industrial and domestic sewage outlets exist along the Shiliu River, and untreated wastewater is directly discharged into the river, causing the water body to turn black and emit foul odors, with serious exceedances of indicators such as COD and ammonia nitrogen (Liu & Xue, 2003; He & Zhang, 2018). The ecosystem in the waters near the sewage outlets has been damaged, aquatic biodiversity has sharply decreased, the sediment appears black and viscous due to heavy metal deposition, and shore vegetation has withered due to sewage infiltration, forming a “sewage-soil-vegetation” chain pollution effect (Yang, Kong & Zhou, 2019). If contaminated river water is used for irrigation in surrounding farmland, it may lead to heavy metal accumulation in crops, further threatening regional ecological security and residents’ health (Wei et al., 2025). This study conducted a comprehensive exploration of the microorganisms present in the sediment and overlying water of the heavily polluted areas of the Shiliu River, providing significant evidence for understanding the river’s microbial ecology. However, it also has certain limitations, and subsequent studies can build upon this foundation. In terms of microbial diversity, the richness of bacteria and fungi in the sediment of the Shiliu River was significantly higher than that in the overlying water. This phenomenon aligns with the findings of most urban river sudies (Huang et al., 2021b). Sediment offers a more complex habitat and abundant nutrient sources for microorganisms, acting like a “microbial reservoir” (Yu et al., 2023; Zhou et al., 2024). Concerning microbial community composition, Proteobacteria was the dominant phylum in both the sediment and overlying water bacteria, although there were notable differences in its relative abundance (Meziti et al., 2016; Zhao et al., 2022; Gao et al., 2024). This dominance aligns with previous urban river studies, but its relative abundance varies across environmental media due to pollution types, levels, and geographical factors (Yu et al., 2025; Chen et al., 2025). For fungi, Rozellomycota was dominant in both the sediment and overlying water fungal samples albeit with differing relative abundances. This may be closely related to its unique ecological adaptability, which enables it to thrive and reproduce effectively in both aquatic and sedimentary environments (Ogaki et al., 2021).

Functional gene analysis revealed significant differences in the metabolic pathways of bacteria and fungi between the sediment and the overlying water of the Shiliu River, reflecting the distinct ecological functions of these microorganism (Djemiel et al., 2022). Among the 25 KEGG metabolic pathways identified in bacteria, the functional gene abundance associated with amino acid metabolism was the highest in both the sediment and overlying water samples. However, the most significant difference was observed in the terpenoids and polyketides metabolic pathways. This divergence can be attributed to the distinct chemical compositions and redox conditions present in the sediment and overlying water, which in turn, influence the activity of microorganisms involved in these metabolic pathways (Zumft, 1997). For instance, the nitrogen cycle, while a natural ecological process, exhibited altered pathway abundances in polluted Shiliu River sediments and overlying water. Bacteria in sediments showed enhanced activity in ammonification pathways, likely driven by elevated organic nitrogen inputs from domestic sewage (Fig. 8). This is supported by the higher abundance of Proteobacteria (32.38% in sediments vs. 56.90% in overlying water), a phylum known to include ammonia-oxidizing bacteria responsive to nutrient pollution. Conversely, overlying water bacteria displayed stronger nitrification potential, which may reflect oxygen-dependent metabolic shifts under polluted conditions. These findings align with studies showing that urban sewage inputs disrupt natural nitrogen cycling by favoring pollution-tolerant taxa and altering pathway fluxes (Reyes et al., 2017; Wan et al., 2023). Additionally, fungi influence material cycling and energy flow through their interactions with dissolved organic matter (DOM). Their community diversity and composition are significantly correlated with high-molecular-weight fluorescent components in DOM, such as fulvic-like and humic-like acids, participating in the formation and transformation of terrestrial-derived DOM (Li et al., 2024). Fungal community characteristics vary significantly among different wetlands. In artificial wetlands like fish ponds, fungal community richness and diversity decrease, altering the transformation of DOM and thereby affecting the material cycling process of ecosystems (Shi et al., 2025). In this study, significant differences in the relative abundances of metabolic pathways associated with g_unclassified_Rozellomycota and g_unclassified_Fungi were observed between the sediment and overlying water samples. This implies that fungi contribute to material cycling and energy flow in various ways depending on environmental conditions, which may be linked to their nutrient-acquisition strategies and differing responses to environmental factors (Arias-Real et al., 2023).

The findings of this study have substantial implications for the protection and restoration of the Shiliu River ecosystem. Based on the observed in microbial diversity and functions, targeted ecological restoration strategies can be developed. In sediment zones with high microbial richness (e.g., dominated by Proteobacteria and Rozellomycota) and functional advantages in amino acid metabolism and nitrogen cycling, in-situ bioremediation techniques such as bioaugmentation (introducing functional strains for pollutant degradation) and biostimulation (adding carbon sources or electron acceptors) can be applied. These methods leverage native microbial communities to enhance the degradation of organic pollutants and nitrogen compounds, as evidenced by the high abundance of nitrogen metabolism genes (nifH, nirS) in sediments (Fig. 8). For the overlying water, where microbial diversity is lower and dominated by Acinetobacter and unclassified_Fungi, specific measures include: (1) enhancing water flow through artificial aeration or channel dredging to increase (2) regulating nutrient input (e.g., reducing phosphorus and nitrogen loads) to suppress harmful algal blooms and favor functional guilds like “Animal Endosymbiont-Plant Pathogen-Undefined Saprotroph” (Fig. 9). These interventions aim to optimize microbial community structure, strengthen carbon and nitrogen cycling, and restore the ecosystem’s self-purification capacity, aligning with the functional differentiation observed between sediment and water columns (Arias-Real et al., 2023).

However, this study has certain limitations. The samples were exclusively collected from the heavily polluted areas of the Shiliu River, which may not comprehensively represent the microbial community characteristics of the entire river. Additionally, the lack of multi-gradient sampling (e.g., lightly polluted or pristine sections) may obscure microbial responses to varying pollution intensities. Studies have shown that microbial community structures and functional potentials differ significantly across pollution gradients (Li et al., 2021). For example, Xiao et al. (2023) revealed distinct nitrogen metabolic pathways in sediments under different pollution levels, highlighting the necessity of gradient sampling for comprehensive ecosystem assessment. This research primarily relied on high-throughput sequencing technology and functional prediction analysis. Further experimental studies are necessary to verify the actual functions of microorganisms, as discrepancies may arise between the predicted and actual functions. Additionally, this study did not consider the influence of factors such as seasonal changes and fluctuations in hydrological conditions on the microbial community. These factors can significantly affect the growth, reproduction, and community structure of microorganisms.

Future research should expand the sampling scope to include samples from various regions of the river and across different seasons. The integration of multi-omics technologies such as metagenomics and metatranscriptomics, will facilitate in-depth investigations into the functions of microorganisms. Furthermore, it is essential to enhance on-site experiments and long-term monitoring. This approach will lead to a more comprehensive and profound understanding of the microbial community in the Shiliu River, thereby providing robust scientific support for its ecological protection and restoration.

Conclusions

This study systematically analyzes the diversity and functions of microorganisms in the overlying water and sediment of the heavily polluted areas of the Shiliu River in Tangshan City, Hebei Province. The research reveals that the richness of bacteria and fungi in the sediment was higher than that in the overlying water. Notable differences exist in the types of bacterial and fungal communities between the two environments, with distinct dominant taxa identified at both the phylum and genus levels. KEGG and COG functional annotations indicated that there were significant differences in multiple metabolic pathways among bacteria, with varying gene abundances for specific functions in the two sample groups. FUNGuild functional predictions also demonstrate variability in the abundances of fungal functional groups between the sediment and overlying water. These findings provide crucial microbiological evidence for the protection and restoration of the Shiliu River ecosystem, while also enriching the research on the microbial diversity in urban river sewage. However, this study has some limitations, including restricted sampling areas, a lack of functional verification, and the omission of environmental factor influences. Future studies should expand the sampling scope, integrate multi-omics technologies to explore microbial functions in greater depth, and strengthen long-term monitoring efforts. Such approaches will facilitate a more comprehensive understanding of the microbial community in the Shiliu River, thereby offering stronger support for its ecological protection and restoration.

Supplemental Information

Supplemental Information 1 PCA.

Supplemental Information 2 NMDS.

Supplemental Information 3 Hclust.

Supplemental Information 4 Barplot.

Supplemental Information 5 Raw data of bacterial Operational Taxonomic Units (OTUs) in sediments and overlying water.

Supplemental Information 6 Raw data of fungal Operational Taxonomic Units (OTUs) in sediments and overlying water.

Supplemental Information 7 Raw data of KEGG function prediction for bacteria in sediments and overlying water.

Supplemental Information 8 Raw data of COG function prediction for bacteria in sediments and overlying water.

Supplemental Information 9 Raw data of fungal function prediction in sediments and overlying water.

Supplemental Information 10 Supplementary Tables.

We are grateful to all lab members in Hebei Key Laboratory of Plant Biotechnology Research and Application for helpful discussions and suggestions in this work.

Additional Information and Declarations

Competing Interests

Qiaoping Sun and Yanfang Zhang are employed by Hebei Green Dream Environmental Science & Technology Co., Ltd.

Author Contributions

Yazi Li conceived and designed the experiments, performed the experiments, prepared figures and/or tables, authored or reviewed drafts of the article, and approved the final draft.

Shuhong Zhang conceived and designed the experiments, authored or reviewed drafts of the article, and approved the final draft.

Yumei Guo performed the experiments, authored or reviewed drafts of the article, and approved the final draft.

Ke Xu performed the experiments, authored or reviewed drafts of the article, and approved the final draft.

Xiqing Zhang conceived and designed the experiments, analyzed the data, authored or reviewed drafts of the article, and approved the final draft.

Mengfei Pan analyzed the data, authored or reviewed drafts of the article, and approved the final draft.

Qiaoping Sun analyzed the data, authored or reviewed drafts of the article, and approved the final draft.

Yanfang Zhang performed the experiments, prepared figures and/or tables, authored or reviewed drafts of the article, and approved the final draft.

Yongshan Fan conceived and designed the experiments, prepared figures and/or tables, authored or reviewed drafts of the article, and approved the final draft.

DNA Deposition

The following information was supplied regarding the deposition of DNA sequences:

The raw sequence reads are available at Genbank: PRJNA1251787 and PRJNA1253430.

Data Availability

The following information was supplied regarding data availability:

The R code is available in the Supplemental File.

The diversity data of bacteria and fungi in sediments and overlying water are available at Sequence Read Archive (SRA): PRJNA1251787 and PRJNA1253430.

The source code of the barplot function in the R base package is available at GitHub: https://github.com/wch/r-source/blob/trunk/src/library/graphics/R/barplot.R.

The source code of the vegan package is available at GitHub: https://github.com/vegandevs/vegan.

The source code of the ape package is available at GitHub: https://github.com/emmanuelparadis/ape.

The source code of the ggpubr package is available at GitHub: https://github.com/kassambara/ggpubr.

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
