# Peer review of "Analysis of microbial diversity and functions in sediments and overlying water of the Shiliu River"

_PeerJ, doi:10.7717/peerj.19979_

## Round 0.1 · original submission · Major Revisions

Dear Dr. Li
You can find the comments and suggestions of the expert reviewers in the attached reports. As you will see, expert reviewers have pointed out the critical errors. Consequently, a major revision is needed for your article.

I request that you improve your manuscript following the reviewers' suggestions

Sincerely

**Language Note:** The review process has identified that the English language must be improved. PeerJ can provide language editing services - please contact us at [email protected] for pricing (be sure to provide your manuscript number and title). Alternatively, you should make your own arrangements to improve the language quality and provide details in your response letter. – PeerJ Staff

Reviewer 1 ·

Basic reporting

This research provides a glimpse into the microbial ecology of the polluted Shiliu River in Tangshan City, China. Samples were collected from the sediment and water column and amplified for the 16S rRNA gene and ITS to understand the diversity of bacteria and fungi. Overall, the manuscript is well written, and the results are well presented and discussed. Some extra information is required in the methodology section for reproducibility reasons, and changes are required throughout to improve the manuscript’s readability.

Please find my comments below.
Title: River needs to be capitalised.

Abstract
Line 23: "Tangshan City, China”. Not everyone knows where Tangshan City is. I would recommend adding this once or twice within the main text as well, specifically line 90 in your methods and line 315 in your conclusion.
Line 30 and throughout: Please change all mentions of “16S rDNA” to 16S rRNA.
Line 53: “The dominant species vary at the level of both the phylum and genus”, but the same fungi taxa are found between water and sediment samples, and the bacteria have the same phyla between samples. This needs to be rephrased as the taxa are the same, but the relative abundance differs.
Line 53 and throughout: “species” is mentioned incorrectly throughout the text. For example, at the phylum level, you find phyla, not species (e.g., line 188). Similarly, at the genus level, you find genera, not species (e.g., line 202). This should be changed to phyla, genera, or taxa where appropriate.
Line 47: “provides a microbiological foundation.” This is too vague. Please expand or rephrase.
Lines 48-49 “enhances the research on microbial diversity in urban river sewage.” This part needs to be expanded more in the discussions because it sounds like there's not enough research on these rivers. Do you mean specifically the Shiliu River?
Line 53: “These results support Shiliu River ecosystem protection.” Too vague, how does it do that?

Introduction
Lines 64-65 “influencing the ecological security of the surrounding areas and the quality of life for residents.” Can you include some examples here? I think it would strengthen your point.
Lines 81-83: Since there is research on the water quality of the Shiliu River, your discussion can be expanded to include some of it. That would bring everything together better and give a better glimpse into the River’s microbial ecology.

Experimental design

Material and methods
Line 98 “sediments samples (B1, B2, B3) and overlying water samples (D1, D2, D3)”: This needs to be included in all figure and table legends, including supplementary materials. It's not a self-explanatory abbreviation, so it must be included everywhere for clarity.

Line 99 “heavily polluted sections of the Shiliu River”: How are these heavily polluted? Do you know from previous research, in which case, can you add references?

Lines 99-102: How was the sediment collected?

Line 105: Do these primers amplify archaea as well? If yes, out of curiosity, did you look into it?

Lines 106-109: You need to add references for these primers.

Line 101 “All samples were mixed in equal amounts at a ratio of 1:1”: What does this mean? Were the samples normalised? 1:1 ratio according to what? Volume, weight, concentration?

Line 119 “barcode tag sequences”: Did you add the tags or Sangon Biotech? Add who did it somewhere (probably in the previous paragraph) and how it was done if you did it.

SILVA and UNITE databases: Please include what versions of these databases you used.

Line 125-126 “R language tools…”: Which ones? versions? R version? Does this include the tools you used for the PCoA and UPGMA? This section needs expanding for reproducibility.

Validity of the findings

Results
Line 137 “valid data”: This isn’t required. If it’s not valid data, just don’t include it.
Line 144 “260,150” (similarly for line 154 for the fungi): How are there more after clustering? Surely it should go down? Or have you now included the sum from all six samples? I think this is confusing and shouldn’t be included. Just say the sediment had X OTUs and the water Y OTUs.
Line 146-148 “features” and again for the fungi (lines 156-158): You need to explain this more. Currently, it reads like there are more OTUs in the water, but here you say the opposite. Where's the data to support this? Also, what are “characteristic sequences”? Unique sequences? OTUs? I think the constant changes in terminology are confusing.
Throughout results: if something is significant, please include the p-values in parentheses (e.g., lines 163 and 170).
Lines 165 and 172 “dilution curves”: Do you mean rarefaction curves?
Line 175 PCA with Bray-Curtis: A PCA uses Euclidean distance between the features. If you’ve used a distance matrix like Bray-Curtis, then you are actually presenting a Principal Coordinates analysis (PCoA).
Line 185 “conversely, the greater the difference”: Do you mean smaller difference? Shorter branch length, more similar, so they are less different?
Lines 189-190: I don't think this sentence is needed—jump straight into it, for example, Proteobacteria have X% relative abundance (Fig. 5A). The same is true for lines 195-196. Overall, your results for bacteria and fungi are written exactly the same, so it would be easier for the reader if these were rephrased slightly to avoid all this repetition.
Lines 191-193: The relative abundance you present here is an average, right? You should include the standard deviation here (+/-)
Line 194 “were selected”: What do you mean they were selected?
Lines 216-220: This whole paragraph needs to go to the materials and methods section.
Lines 222: Include here what the p-value threshold was; “lowest p-value” is a bit vague.
Line 229 “encompassed”: I’m unsure why that word is there.
Line 230-231: How? This is very quickly concluded. Include and expand on this in the discussion section. Why can't these just be naturally there? It sounds like they are associated with pollution only - all of these are natural processes.
Lines 235-236 “[R] General function prediction”: You also mention this in your abstract. What does it mean? Give some examples.
Lines 239-241: This sentence should be moved to the materials and methods section.

Discussion
Overall, your discussion was very well developed and written.
Lines 258-259: This sentence needs a couple of references.
Lines 281-284: Can you give a few examples here? It would strengthen your comment.
Lines 287-290: This sounds a bit vague. Can you expand?
Figures and tables
Please include what B and D stand for in all figure and table legends, including supplementary ones.
Figure 1: Please include what the numbers within the circles are in the legend. OTUs? Sequences? Phyla? Genera?
Figure 3: See my comment on PCA vs PCoA, and include that you used Bray-Curtis in your legend.
Figure 8: You don't need the colour to differentiate between B and D, as they appear as labels under the heatmap. The blue and red of B and D overcomplicate the heatmap, as the colours for abundance are the same, so you may as well remove them.
Figure 9 “The horizontal axis represents the samples, and the vertical axis represents the abundance proportion of the guild in different samples”: I don’t think this explanation is required.
Tables 1 and 2: It might be easier to read these tables if you add SD as a +/—value after the average rather than in a different row.
Supplementary Excel spreadsheets: These tables need legends to explain what they are.

Reviewer 2 ·

Basic reporting

1. I am not a native English speaker, so I cannot comment on the English proficiency of the paper.

2. The literature references can be improved/enriched.

3. attracting considerable attention regarding their ecological status and biodiversity. PLEASE CLARIFY, ECOLOGICAL STATUS INCLUDES BIODIVERSITY ELEMENTS ....

4. Relevant self-contained results to the hypothesis.

5. sediment and overlying water
Samples were collected from its severely polluted reaches. WHY NOT LESS POLLUTED AND NOT POLLUTED REACHES FOR COMPARISON?

6. The results indicate that the richness of bacteria and fungi in the sediments is greater than that in the overlying water. STATE THAT THIS IS A NORMAL SITUATION

7. This study provides a microbiological foundation for the protection and restoration of the Shiliu River's ecological system and enhances the research on microbial diversity in urban river sewage. ONLY FOR THIS?

8. IN MY OPINION, THE APPROACH IS TOO DESCRIPTIVE IN TERMS OF THE STRUCTURE OF BIOTA AND TOO LESS IN TERMS OF THEIR FUNCTIONS

9. Rivers are a vital component of the ecosystem, playing a crucial role in maintaining regional
58 ecological balance, providing water resources, and supporting biodiversity. ONLY?

10. Microorganisms, as essential members of the river ecosystem, are indispensable in various
60 ecological processes, including material cycling, energy flow, and pollutant degradation ONLY?

11. The microbial communities present in the sediments and overlying water not only 62 reflect the health of the river ecosystem but also are crucial for the proper functioning of its
63 ecological processes. WHICH PROCESSES?

12. ALL OF THE NON-ORIGINAL TEXTS/SENTENCES/PARAGRAPHS SHOULD HAVE CITATIONS AND REFERENCES.

13. Understanding the microbial 68 diversity and functions of the sediments and overlying water in the Shiliu River is vital for PeerJ reviewing PDF | (2025:04:117596:0:1:NEW 7 May 2025) Manuscript to be reviewed 69 elucidating the operational mechanisms of the river ecosystem, evaluating its ecological health 70 status, and formulating targeted ecological restoration strategies. ONLY?

14. Analyzing microbial diversity allows for a better understanding of ecosystems, 73 responses to environmental changes ONLY?

15. Furthermore, the functions of microorganisms, particularly
77, Those involved in the cycling of essential elements such as carbon, nitrogen, and phosphorus are 78, crucial for the normal operation of river ecosystems ONLY?

16. and providing a scientific basis for the 86 protection and restoration of the Shiliu River ecosystem. ONLY?

17. YOUR RESULTS ARE IMPORTANT AND CORRECTLY OBTAINED, BUT NOT ENOUGH DEEP AND COMPLEX PERSPECTIVE DIMINISHES THEIR VALUE CONCERNING THE POSSIBLE USE IN THE FUTURE OF ECOLOGICAL-RELATED POTENTIAL GOALS. COAUTHORSHIP WITH AN ECOLOGIST CAN HIGHLIGHT YOUR RESULTS AND PUT THE PAPER RESULTS IN A MUCH BETTER PERSPECTIVE FOR THE READERS. IN OTHER WORDS, IT IS DESCRIPTIVE ENOUGH BUT LESS EXPLANATORY.

18. WHEN/IF HIGHLIGHT THE LIMITATIONS OF YOUR WORK APPROACH, PERSPECTIVE, RESULTS, AND CONCLUSIONS, ADDING WHICH IS STILL THE IMPORTANCE OF YOUR APPROACH

19. This suggests that
261, The pollution status and geographical conditions of the Shiliu River have a significant impact on 262, the bacterial community structure. PLEASE EXPLAIN

20. THE DISCUSSIONS ARE IN THE MAJORITY IN A COMPARATIVE WAY BETWEEN WATER AND SEDIMENT, BUT ARE LESS AMONG THE DIFFERENT STUDIED REACHES

21. Based on the observed microbial diversity and functions, targeted 287 ecological restoration strategies can be developed. MORE PROPOSALS HERE?

Experimental design

Original primary research within the Aims and Scope of the journal. OK

Research question OK

Rigorous investigation OK

Methods described OK

Validity of the findings

Novelty OK. Impact RELATIVELY LOW

Additional comments

CONGRATULATIONS ON YOUR WORK AND SUCCESS! REVIEWER

---

## Round 0.2 · accepted · Accept

Dear Dr. Li,

I thank you for making the corrections and changes requested by the reviewers. I read and checked your valuable article carefully and am happy to inform you that the article has been accepted for publication in PeerJ.

I kindly request you to take into account reviewer 1's minor correction in the final proof of your manuscript.

Sincerely yours,

Reviewer 1 ·

Basic reporting

This research provides a glimpse into the microbial ecology of the polluted Shiliu River in Tangshan City, China. Samples were collected from the sediment and water column and amplified for the 16S rRNA gene and ITS to understand the diversity of bacteria and fungi. Overall, the manuscript is well-developed and well-written, and I believe the authors addressed the reviewers’ concerns adequately, thus considerably improving it.

Experimental design

I only have one minor comment:
Line 125-126: “All samples were mixed in equal amounts at a ratio of 1:1.” Once again, this doesn’t include what “amounts” means. From the comment you left to the reviewers, I take it to mean mixed according to moles, so I recommend writing something like: “All samples were pooled at equimolar ratios.”

Validity of the findings

No comment

Additional comments

No comment

Reviewer 2 ·

Basic reporting

no comment

Experimental design

no comment

Validity of the findings

no comment

Additional comments

no comment